# Traceability Management System Using Blockchain Technology and Cost Estimation in the Metrology Field

**DOI:** 10.3390/s23031673

**Published:** 2023-02-03

**Authors:** Naoki Takegawa, Noriyuki Furuichi

**Affiliations:** National Metrology Institute of Japan (NMIJ), National Institute of Advanced Industrial Science and Technology (AIST), 1-1-1, Umezono, Tsukuba 305-8563, Ibaraki, Japan

**Keywords:** measurement, traceability, management, blockchain, operating cost

## Abstract

Metrological traceability is essential to ensure the reliability of calibration tests. Calibration certificates usually include information on only one upper-level reference standard. As metrological traceability is multi-layered, generally there is no method available for end users to instantly confirm the traceability from the reference standard to a primary standard. This study focuses on the Ethereum blockchain, which has both tamper resistance and high availability, as a digital data management method. To improve the transparency and reliability of calibration tests, a smart contract that traces back to the primary standard is proposed. Consequently, it is confirmed that end users can instantly obtain traceability information. In addition, the execution of smart contracts requires transaction fees. Here, the calculation of the transaction fees is organized, and the traceability management system is discussed from a cost-effective perspective in the field of metrology.

## 1. Introduction

The introduction describes metrological traceability and examines the benefits of its management and visualization system. It also surveys the previous literature on digital calibration certificate (DCC) and the use of blockchain, which are related technologies for digitizing metrological traceability. The overview of this research and the contents of the chapters are provided.

### 1.1. Issues Related to Traceability Management in the Field of Metrology

In manufacturing, accurate measurement is indispensable for achieving the required quality and improving productivity. Calibration of equipment with reference standards is important in ensuring the reliability of measurement results, and the establishment of metrological traceability is required in international standards, such as ISO 10012 [1] and ISO 17025 [2]. Metrological traceability is defined in ISO/IEC Guide 99:2012 VIM [3] as “Property of a measurement result whereby the result can be related to a reference through a documented unbroken chain of calibrations, each contributing to the measurement uncertainty”. A calibration certificate issued by an accredited calibration laboratory is a stand-alone proof of metrological traceability and usually includes information on only one upper-level reference standard. Therefore, as metrological traceability is multi-layered, there is no immediate way for end users to ascertain the traceability from a device under test (DUT) to a primary standard in general. Note that the results in a certificate are metrologically traceable in the above situation. Takatsuji et al. [4] highlighted the existence of certificate holders who require details of metrological traceability and proposed the visualization of metrological traceability. Miličević et al. [5] presented the concept of a traceability system for electrical energy measurement based on blockchains. Although the management and visualization system of metrological traceability is important and expected to improve the reliability and transparency of calibration tests, there are several issues related to (1) the digital format and (2) the digital security of calibration information in its construction.

### 1.2. Calibration Information as Digital Data and Digital Calibration Certificate (DCC)

To organize the handling of calibration information as digital data, studies on DCC have been conducted [6,7,8]. Hackel et al. [9] mentioned the necessary content for DCC and proposed XML as a data format. Mustapää et al. [10] discussed the contribution of DCC to the uncertainty, completeness, and authenticity of measurement data and introduced applications of DCC such as smart cities. Ačko et al. [11] summarized the SmartCom (communication and validation of smart data in IoT networks) project adopted by EMPIR of EURAMET, which aims to develop a digital format, and presented the interim progress of SmartCom, including the DCC format. Gadelrab et al. [12] conducted an exhaustive survey study on DCC. In addition, CIPM is actively undertaking a project [13] called Digital-SI to establish a metadata format that conforms to SI units.

### 1.3. Digital Security of Calibration Information and Blockchain Technology

In recent years, the use of blockchain as a highly robust and highly available database in the industrial sector has increased [14,15,16,17,18]. Compared with the usual centralized databases, the advantages of blockchains are considered to be the tamper resistance, elimination of single points of failure, high availability, and savings in human costs. Andoni et al. [19] reviewed the current business cases and presented blockchain solutions for the energy industry. Zhou et al. [20] summarized the global development of peer-to-peer energy trading and introduced blockchains supporting the trading. Leng et al. [21] investigated the contribution of blockchains to achieving sustainability from the perspective of the manufacturing system and product lifecycle management. Chen et al. [22] designed an efficient and secure data collection framework in the smart grid by integrating fog computing and blockchain. Iftikhar et al. [23] studied recent research based on blockchains in the IoT for privacy protection. Suvarna et al. [24] focused on the concept of cyber–physical production systems (CPPSs) and proposed applying blockchains to CPPS to secure data sharing in decentralized systems.

In metrological transactions, the tamper resistance of calibration information is important because there is an economic incentive to tamper with the information on measuring instruments. When traceability management and visualization are implemented as a system, high system availability and elimination of single points of failure are also required. Therefore, blockchain has attracted attention as a useful tool in the field of metrology [25,26,27]. Blockchains can be broadly classified into private and public chains based on the presence or absence of an administrator. Table 1 presents a summary of private and public blockchains. Hyperledger Fabric (HF) [28] is a private chain that is being considered for use in metrology. Moni et al. [29] used HF to connect peers between the national metrology institute in Germany and Brazil to identify and authenticate smart meters. Melo et al. [30,31] compared the blockchain with existing paper- and cloud-based data management and examined the throughput and latency of HF applied to smart meters. Yurchenko et al. [32] proposed a model for a secure smart meter system using HF and cryptography. Peters et al. [33,34] proposed a use of blockchain in legal metrology and verified the confidentiality of decentralized meters combining HF and homomorphic cryptography.

One public chain that is being considered for use the field of metrology is Ethereum [35,36]. Gavin [37] released the first yellow paper outlining the technical specifications of Ethereum. Ethereum is explained in detail in Section 2 and beyond. Iqbal et al. [38] addressed issues related to trust in IoT systems and proposed tracking, management, governance, and access control of smart vehicles using Ethereum. Shah et al. [39] proposed the management of calibration information using Ethereum and examined the effect of the number of calibrators and calibration hierarchy on the time to obtain traceability in the Ethereum blockchain. Santis et al. [40] proposed a combination of blockchain and physical unclonable function-based authentication protocols for an auditing system for metrological traceability. The system for voltage and current measurements used Ethereum as the blockchain technology and Node.js as the web interface. Peterek and Montavon [41] proposed the IOTA [42,43] blockchain, a public chain, as a database of hash values of measured data.

**Table 1 sensors-23-01673-t001:** Comparison of private and public chains.

	Private Chains	Public Chains
Consensus building	Consensus-building costs (fees and time) are generally small.	Consensus-building costs (fees and time) are generally significant.
Robustness	The possibility exists that data may be tampered with by certain participants. A single point of failure may exist.	For the cost of consensus building, the likelihood of data being falsified by a particular participant is generally low.
Chain participants	Specific (licensed individuals and companies)	Unspecified
Application examples in the field of metrology	HF [26,30,31,32,33,34,44]	Ethereum [26,38,39,40,45,46,47], IOTA [41]

### 1.4. Contents of This Study

As described in Section 1.1, the management and visualization system for metrological traceability in this study is expected to improve the reliability and transparency of calibration tests. Therefore, the above system is constructed using blockchain, which has been attracting attention in the field of metrology. The digital format of calibration information is beyond the scope of this study, as there are numerous studies and projects on this topic. An important concept in this study is the recognizable traceability path (RTP), which is defined as the path from a DUT to a primary standard that can be recognized by end users (Figure 1). In Figure 1, the traceability hierarchy is set to 4 as an example. However, in practice there are simpler or more complex cases than this case. Additionally, examples of standards in the flow measurement field corresponding to each hierarchy are provided. By using the RTP, end users can easily know metrological traceability. Although the RTP mainly focuses on information in calibration certificates that is metrologically traceable, a system such as the RTP can also be explored for other information that requires traceability management and visualization.

Section 1 summarizes the management and visualization system of metrological traceability and the adoption of blockchains in the field of metrology. In Section 2, a simple system for the RTP is created using the Ethereum blockchain, which is extremely difficult to tamper with. There are only a few studies in the field of metrology that have written a blockchain program and verified its operation. Section 3 estimates the cost of recording information on the Ethereum blockchain and examines the Ethereum-based traceability management and visualization system from an economic cost perspective. There is a debate about whether to choose private or public chains, such as HF or Ethereum, respectively. An important indicator for choosing between private or public chains is the cost of the chain (Table 1). However, to the best of the authors’ knowledge, there is no study in the field of metrology that examines the economic feasibility of using blockchain to manage digital data.

## 2. Building a Traceability Management System Using a Smart Contract

### 2.1. Smart Contract of Ethereum

The Ethereum blockchain has a feature called a smart contract [48,49] that automatically executes contracts (programs), allowing for complex processing. Applications using smart contracts are examined in detail by Hewa et al. [50]. For instance, smart contracts have been used to develop various services, such as decentralized finance (DeFi) and non-fungible token [51], which is proof of the uniqueness of digital art. Smart contracts are often written in a programming language called Solidity [35]. The smart contract on the RTP described below is written in Solidity and their behavior is checked on Ropsten, Ethereum’s test network. The behavior is also confirmed on another test network, Goerli.

### 2.2. Preparations Required to Execute Smart Contract

There are many methods for deploying and using smart contracts written in Solidity on the Ethereum network. One method is to use Go Ethereum (Geth), a node operation software developed by the Ethereum Foundation, and another method is to use an integrated development environment called Remix, which allows the creation, compilation, and deployment of smart contracts in a web browser. Unlike the above methods, this study implements and uses smart contracts, employing Truffle, INFURA, and web3.js, which are relatively easy and highly flexible. Truffle is the de facto standard framework for Ethereum application development and can compile and deploy smart contracts. INFURA, an Ethereum node hosting service, makes it possible to connect to the Ethereum network without downloading Ethereum nodes such as Geth. web3.js is a JavaScript library and can be used to access deployed smart contracts. Figure 2 illustrates these relationships.

### 2.3. Smart Contract on the RTP

An overview and program of a simple smart contract on the RTP are depicted in Figure 3 and Algorithm 1. There are two types of functions in Algorithm 1. One is “function add” that records data such as a DUT or reference standard in the Ethereum blockchain, and the other is “function RTP” that references the RTP from a DUT to a primary standard. “constructor()” is executed when a smart contract is deployed and records the address of the smart contract issuer. This grants access to “function add”, which is described next only to the smart contract issuer. This is necessary to resolve user authentication issues such as verifying the identity of national metrology institutes and calibration laboratories.

“function add” records the two arguments, “string data” and “uint referenceID”, on the Ethereum blockchain. “referenceID” indicates the ID with which the information on one upper-level reference standard is linked. Transaction fees must be paid when using “function add” because of the data writing process involved. According to Hackel et al. [9], there are six types of information to be recorded as string data: identifier, measurement value, expanded measurement uncertainty, coverage factor, unit, and time. If six items are recorded as RTP data, it is desirable to prepare six types of variables (“string data”) to store data. The data recorded in the smart contract should be limited to what can be disclosed to the outside world, and information that cannot be disclosed should be recorded as hash values. “function RTP” allows the end user to enter the ID of a DUT and receive RTP information as the return value. In the case of data for reference only, no transaction fees are incurred. To make this clear, the function modifier “view” is used in “function RTP”.
**Algorithm 1:** Smart contract on the recognizable traceability path (RTP) written using Solidity in this study.1:2:3:4:5:6:7:8:9:10:11:12:13:14:15:16:17:18:19:20:21:22:23:24:25:26:27:28:29:30:31:32:33:34:35:36:37:38:39:40:// SPDX-License-Identifier: MITpragma solidity ^0.8.0;contract RecognizableTraceabilityPath {   address public institutionID;   Point[] public points;   struct Point {     string data;     uint referenceID;  }   constructor() {     institutionID = msg.sender;  }   function add(string memory _data, uint _referenceID) public {     if(institutionID != msg.sender){      revert();     }else{       points.push(Point(_data, _referenceID));     }  }   function RTP(uint _startID) view public returns(string[] memory) {     uint ID = points[_startID].referenceID;     uint rank = 1;     while(ID != 0) {       rank++;      ID = points[ID].referenceID;     }     string[] memory rtp = new string[](rank);     rtp[0] = points[_startID].data;     ID = points[_startID].referenceID;     for (uint i = 1; i < rank; i++) {       rtp[i] = points[ID].data;       ID = points[ID].referenceID;     }     return rtp;   }}

In this study, calibration clients and end users are assumed as User 1 using “function RTP”, and national metrology institutes and accredited calibration laboratories are assumed as User 2 using “function add”. An example of RTP information received by User 1 is depicted in Figure 4, revealing the actual acquisition of pre-registered data from Ropsten (Ethereum’s test network) by a local server built on node.js through web3.js and INFURA. The calibration information in Figure 4 corresponds to “string data” in Algorithm 1, and the data are stored in each traceability hierarchy. Information registered on Ropsten using Algorithm 1 is obtained through web3.js and INFURA and reflected in node.js. Traceability hierarchy 1 describes the calibration information on a primary standard, and 4 describes the calibration information on a DUT.

## 3. Cost Estimation of the RTP

### 3.1. Overview of Transaction Fees

Section 3 estimates the costs associated with the RTP proposed above and discusses the economic feasibility of managing calibration information using Ethereum, a public chain. To deploy and execute a smart contract on Ethereum, a transaction must be issued and a fee called “gas” must be paid. The transaction fees are paid to the miner, who tends to record transactions with high transaction fees in the blockchain. This gas-based transaction fee is represented by the following equation:(1)gas(eth)=gas price(eth/gas)×gas usage(gas)

The gas price is set by the transaction issuer based on the congestion of the Ethereum network. The gas usage varies depending on the content of the program to be executed by the smart contract. Therefore, to estimate the cost of a smart contract for the RTP, it is necessary to estimate the gas price and gas usage.

### 3.2. Gas Price

The gas price can be set by the transaction issuer. Setting a higher gas price increases the likelihood that the transaction is recorded in the blockchain more quickly. Therefore, the gas price setting depends on the urgency of transaction approval. The gas price comprises three components: Base Fee Per Gas, Priority Fee Per Gas, and Max Fee Per Gas. For more details, please refer to EIP-1559 [52]. As an indication of the gas price, you can use web3.eth.getGasPrice() of web3.js, a JavaScript library, to obtain the median gas price set in multiple transactions in the past. Figure 5a depicts the time variation in the gas prices obtained from web3.eth.getGasPrice() observed in the main network of Ethereum. The units of gas price are expressed in Gwei (=10^−9^ ETH). The gas price is affected by various factors, such as the congestion of Ethereum’s main network, etc. Therefore, it is extremely difficult to predict future gas prices even though it is possible to know the current appropriate gas prices. It is not necessary to pay for the gas price depicted in Figure 5a at each time, where users can set their gas price as low as not less than Base Fee Per Gas. Figure 5b is the gas price converted to the dollar notation by multiplying Figure 5a with the price of Ethereum, providing an intuitive gas price. The gas prices represented in Figure 5a or Figure 5b multiplied by the gas usage is the transaction fee on Ethereum’s main network.

### 3.3. Gas Usage

Languages such as Solidity that describe smart contracts on Ethereum are compiled into bytecode and opcode that are executed by the Ethereum Virtual Machine in the node. The gas usage is determined by the arithmetic operations performed in the opcode. For example, addition of two elements (ADD) consumes three gas; multiplication of two elements (MUL) consumes five gas; obtaining block height (NUMBER) consumes two gas, and obtaining balance (SELFBALANCE) consumes five gas in the opcode. All the above opcodes are static gas costs; the opcodes whose gas cost varies depending on the amount of data handled are called dynamic gas costs. For more information on each opcode, please refer to the Ethereum Yellow Paper (Berlin) [53]. The gas usage required to execute “function add”, which records the calibration information described in Algorithm 1, can be estimated by examining the opcodes used. In this study, “function add” is executed on Ropsten (Ethereum’s test network) to extract the opcode operations with large gas usage and verified the dominant factors in transaction fees.

The results of the validation reveal that when “string data” and “uint referenceID” are recorded on Ropsten in Algorithm 1, the gas usage is 72,572. The two opcodes with the highest gas costs are TRANSACTION and SSTORE, as depicted in Figure 6. TRANSACTION consumes 21,000 gas as the minimum cost of issuing a transaction. SSTORE consumes 22,100 gas as the cost of recording data (cold access) on each Ethereum node. In the program described here, SSTORE is used twice because two data, “string data” and “uint referenceID”, are recorded (44,200 gas). TRANSACTION and two SSTOREs account for approximately 90% of the total gas usage of 72,572 gas. This ratio is similar when the number of variables (number of “string data”) increases. From this, it is possible to estimate the overall gas usage from the number of times SSTORE are used, i.e., the number of variables. Note that recording more than 256 bits of data in “string data” may cause the gas usage to fluctuate.

### 3.4. Feasibility of RTP Using Ethereum in the Field of Metrology

According to the Physikalisch-Technische Bundesanstalt (PTB) report [9], the number of calibration certificates issued is estimated to be approximately 10,000 per year by PTB and approximately 100,000 per year by accredited calibration laboratories in Germany. In addition, the National Metrology Institute of Japan (NMIJ) issues approximately 700 calibration certificates per year (jcss calibration: 500, calibration and testing service except jcss: 200), excluding those in the field of legal metrology, and approximately 610,000 calibration certificates [54] are issued by accredited calibration laboratories in Japan. As mentioned above, the number of calibration certificates issued by each national metrology institute is enormous, and it is unrealistic to manage all the information in a public chain, which requires transaction fees. For a single measurement, the cost of recording the six items proposed by Hackel et al. [9] on the Ethereum blockchain using the smart contract described in Algorithm 1 is estimated. As the majority of the gas usage is accounted for by one TRANSACTION and six SSTOREs, the usage is estimated to be approximately 170,000 gas based on Section 3.3. The necessary transaction fees for a single measurement can be estimated by multiplying this gas usage by the median gas price of 37 Gwei/gas (Figure 5a) or 0.000089 USD/gas (Figure 5b). For reference, as of 2022, the NMIJ’s fee for issuing a certificate for calibration and testing service except jcss is 1300 JPY for Japanese text and 2300 JPY for English text (excluding tax).

The data recording on a blockchain is expected to improve the reliability and added value of the measurement system, as exemplified by the traceability management and visualization system (i.e., the RTP) proposed in this study. Therefore, companies and research institutions should conduct a cost–benefit analysis before recording data on a blockchain. Blockchain usage may be appropriate for valuable data with high calibration costs (e.g., flow meter calibrations with large bore under high flow rates and calibration of cryogenic thermometers) because of the need to pay transaction fees at the time of using a public chain (Ethereum). Moreover, important standards that are frequently referenced, such as national and primary standards, would also be worth registering on the blockchain. If the transaction fee is an issue in the use of blockchain, utilizing a private chain such as HF rather than a public chain such as Ethereum would reduce costs. However, tamper resistance and other factors need to be considered when using a private chain. On public chains, practical applications for lowering transaction fees are also underway. For instance, Layer2 technologies such as Lightning Network [55,56] and proof-of-stake [57,58] alternatives to proof-of-work [59] are being developed.

## 4. Conclusions

This study focuses on the Ethereum blockchain, which is both tamper-resistant and highly available, as a method of managing digital data in the field of metrology as only a few studies have clearly identified the economic costs of blockchains. This study then proposes the RTP that can be accessed by end users using smart contracts to improve the transparency and reliability of calibration tests. Only a few existing studies have created smart contracts and verified their operation. While describing the development environment and procedures in detail, this study works on the management and visualization of the traceability path. As a result, the recording of data on the blockchain (“function add” in Algorithm 1), the retrieval of data from the blockchain (“function RTP” in Algorithm 1), and the verification of the output as the RTP (Figure 4) are confirmed. Furthermore, using Ethereum, the transaction fee of executing smart contracts is estimated. The calculation of the transaction fee requires gas prices and gas usage. First, the required gas prices are recorded (2022/03~2022/07) and clarified on Ethereum’s main network. Then, calculation method of the gas usage is explained in detail, and the opcodes of TRANSACTION and SSTORE account for 90% of the gas usage in the smart contract for the present RTP. In addition, the traceability management system is verified from an economic cost perspective. The basic cost of executing smart contracts on Ethereum is described so that everyone can reproduce it. This study makes a valuable contribution to the literature by presenting a decision method based on economic costs in an era where there is debate in the field of metrology about choosing between private and public chains such as HF and Ethereum, respectively.

## Figures and Tables

**Figure 1 sensors-23-01673-f001:**
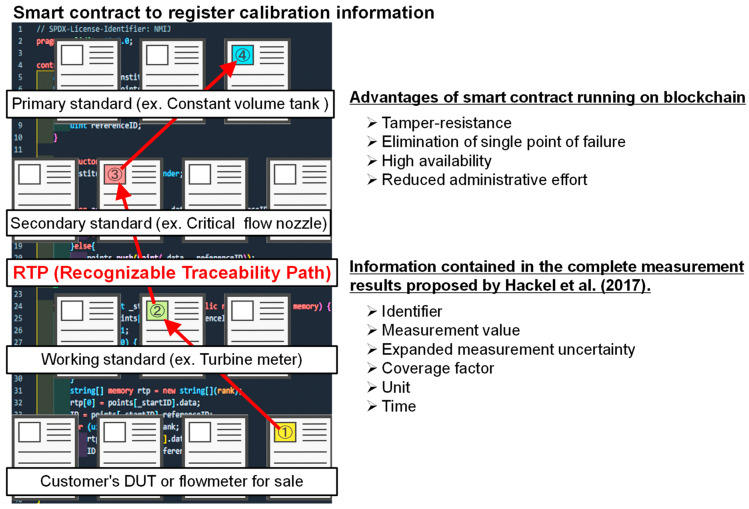
Conceptual diagram of the RTP using a smart contract. Hackel et al. [9] mentioned the necessary content for DCC.

**Figure 2 sensors-23-01673-f002:**
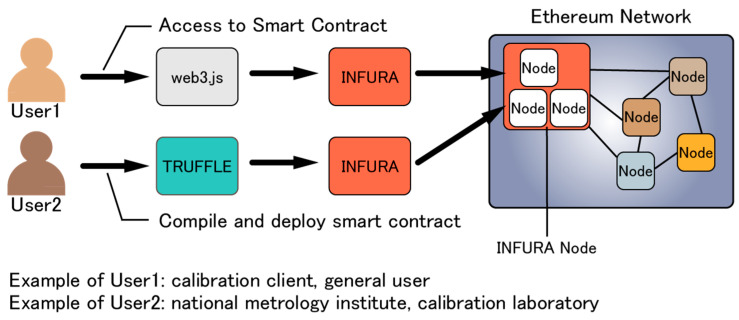
Method of creating and using smart contracts in this study.

**Figure 3 sensors-23-01673-f003:**
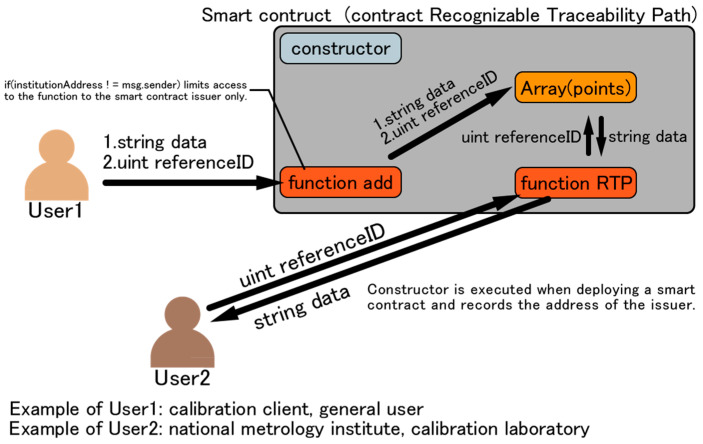
Overview of smart contract on RTP in this study.

**Figure 4 sensors-23-01673-f004:**
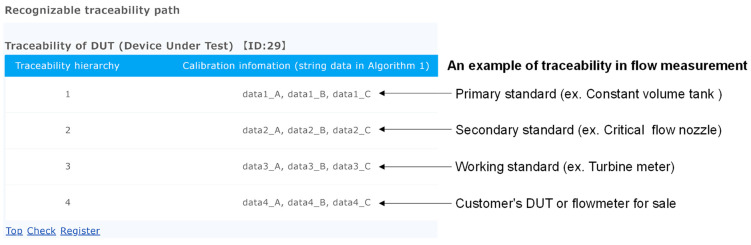
An example of the RTP received by an end user.

**Figure 5 sensors-23-01673-f005:**
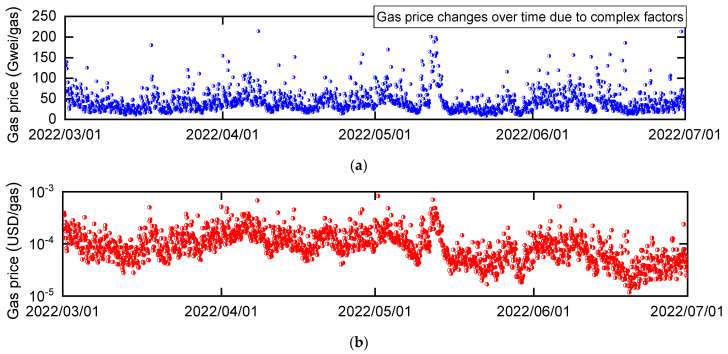
Time variation in gas price on Ethereum’s main network. The transaction fee is determined by the gas price and gas usage. Gas prices change over time due to various factors. Gas usage depends on the throughput of smart contracts (type and number of opcodes executed). (**a**) Gas price (Gwei/gas). (**b**) Gas price (USD/gas).

**Figure 6 sensors-23-01673-f006:**
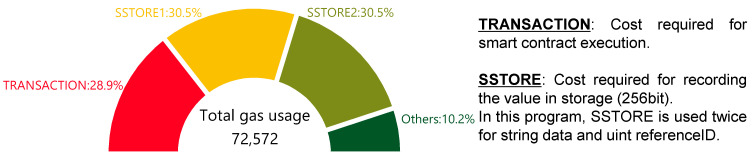
Breakdown of gas usage in a transaction.

## Data Availability

The data that support the findings of this study are available from the corresponding authors upon reasonable request.

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
