# Peer review of "Traceability Management System Using Blockchain Technology and Cost Estimation in the Metrology Field"

_sensors, 2023, doi:10.3390/s23031673_

Round 1

Reviewer 1 Report

1. Consider using "Algorithm" instead of "Algorism".

2. It is not completely clear why the authors are analyzing in detail the Ethereum chain. Namely, it is, of course, possible to use a customized private chain (also Ethereum/EVM based) which does not depend on the Ethereum transaction costs, and in this case, costs could be adapted to a particular use case, i.e. analysis presented in the paper would not be relevant. So, please clarify the relevance of your paper in this context.

3. Authors should clarify how they would solve the problem of user authentication, e.g. confirm the identity of the national metrology institute, and calibration laboratory?

4. Detailed presentation of opcode and bytecode (in Fig. 7) is not needed.

5. Please analyze in more detail why you would use blockchain instead of the usual centralized database.

Author Response

We wish to express our appreciation to the reviewer for his or her insightful comments, which have helped us significantly improve the paper.

1-1)

Comment

Consider using "Algorithm" instead of "Algorism".

Response

We have changed the description to "Algorithm" throughout the manuscript as you indicated.

1-2)

Comment

It is not completely clear why the authors are analyzing in detail the Ethereum chain. Namely, it is, of course, possible to use a customized private chain (also Ethereum/EVM based) which does not depend on the Ethereum transaction costs, and in this case, costs could be adapted to a particular use case, i.e. analysis presented in the paper would not be relevant. So, please clarify the relevance of your paper in this context.

Response

I agree with your point that it is possible to use private chains and adapt costs to specific use cases. In the field of the metrology, Ethereum is widely used for public chains and Hyperledger Fabric (HF) for private chains. And each chain has the advantages and disadvantages (Table 1). However, there are very few papers in the field of metrology that discuss the transaction fee, which is a disadvantage of public chains. Therefore, this paper describes it in detail. On the other hand, this paper does not necessarily recommend public chains or Ethereum. We hope that the clarification of the transaction fee for Ethereum, which is a representative public chain, will allow users to make a rational choice between public and private chains after understanding the advantages and disadvantages.

1-3)

Comment

Authors should clarify how they would solve the problem of user authentication, e.g. confirm the identity of the national metrology institute, and calibration laboratory?

Response

If the problem of the authentication is to be solved on the blockchain, it would be practical to register IDs linked to calibration laboratories on the blockchain in advance, and then use the ID and password to authenticate when using the system. Our program shown in Algorithm 1 is also considered with respect to certification. We have added the above information (L. 161~162).

Registration of authentication ID : line 12~14 in Algorithm 1.

Authentication of system availability by ID: line 17~19 in Algorithm 1.

1-4)

Comment

Detailed presentation of opcode and bytecode (in Fig. 7) is not needed.

Response

As you indicated, we agreed that the bytecode and opcode details were unnecessary. We have therefore removed these.

1-5)

Comment

Please analyze in more detail why you would use blockchain instead of the usual centralized database.

Response

First, the use of blockchain rather than a centralized system is expected to contribute to the prevention of tampering. Secondly, since blockchain is a decentralized system, it is resistant to system failures. This means that the system is fault tolerant. Therefore, a reduction in human management costs can be expected. We have added the above information (L. 60~62).

Reviewer 2 Report

The work is well structured and English language looks quite good. The article is written intelligibly in a good scientific style and is easy to read.

However, there are some issues at the paper, which should be addressed:

1.       In Figure 1, please cite correctly the reference “Hackel et al (2017)”

2.       Figure 1 – name too long. The explanations of Fig.1 can be in the main text.

3.       Don’t use first person pronouns "we".

4.       The conclusion should be extended to specify the received scientific results.

5.       Extend the reference list with relevant publications, published in MDPI journals.

Author Response

We wish to express our appreciation to the reviewer for his or her insightful comments, which have helped us significantly improve the paper.

2-1)

Comment

In Figure 1, please cite correctly the reference “Hackel et al (2017)”

Response

We have sited the reference as "Hackel et al. [9]" in Fig. 1.

2-2)

Comment

Figure 1 – name too long. The explanations of Fig.1 can be in the main text.

Response

As you indicated, we have shortened the caption name for Fig. 1 and moved the specific description to the main text.

2-3)

Comment

Don’t use first person pronouns "we".

Response

The pronouns "we" has been removed from the main text.

2-4)

Comment

The conclusion should be extended to specify the received scientific results.

Response

As you indicated, we have added the scientific results obtained in the conclusion section and explained the results in detail (L. 302~311).

2-5)

Comment

Extend the reference list with relevant publications, published in MDPI journals

Response

We have added six new pertinent papers (References [14] [15] [16] [17] [18] [27]).

Reviewer 3 Report

The authors propose a smart contract that traces back to the primary standard to improve the transparency and reliability of calibration tests. A validation of the proposed recognizable traceability path (RTP) is also done to measure the feasibility of RTP using Ethereum in the field of metrology using smart contracts to improve the transparency and reliability of calibration tests and data.  In general, the paper's technical contributions can be accepted with good presentation style. The organizational structure of the paper is also good. The authors need to proofread some parts. Therefore I recommend minor revision.

Some comments:

- Introduction section can be improved better. It is better to have some introductory paragraphs before subsections.

- The text in Fig. 1 is too large in comparison to text fonts.

- What does "Algorism" mean in text?

- It should be Gas usage in Fig. 5(a) legend.

- Fig. 6 can be put into Appendix since bytecode and opcode seems not interesting to be in main text.

Author Response

We wish to express our appreciation to the reviewer for his or her insightful comments, which have helped us significantly improve the paper.

3-1)

Comment

Introduction section can be improved better. It is better to have some introductory paragraphs before subsections.

Response

As you indicated, we have added the overview of the introduction to the previous paragraph of the subsection to improve the outlook of the introduction (L. 21~25).

3-2)

Comment

The text in Fig. 1 is too large in comparison to text fonts.

Response

The font size in Fig. 1 has been adjusted to the size of the main text.

3-3)

Comment

What does "Algorism" mean in text?

Response

First, we have changed the description to "Algorithm" throughout the manuscript based on suggestions from other reviewers. Second, we believe that "Algorithm" in the text shows a caption (e.g., Figure and Table) to describe a program code. Of course, this notation may vary depending on the research field.

3-4)

Comment

It should be Gas usage in Fig. 5(a) legend.

Response

Fig. 5 (a) and (b) show the time variation of gas prices. We have removed a reference to gas consumption in the legend has been removed because it is misleading.

3-5)

Comment

Fig. 6 can be put into Appendix since bytecode and opcode seems not interesting to be in main text.

Response

The other reviewer also pointed out that the bytecode and opcode details were not necessary, so we have removed the figure showing the bytecode and opcode.

Reviewer 4 Report

The manuscript presents a traceability management system for calibration certificates based on blockchain technology. It is quite well written, with few spelling and grammar mistakes. The relevance is clear and the methods are presented in detail, so they can be reproduced by interested parties. 

I've found only some smaller issues that should be easily addressed, as follows:

- in line 26, VIM 2007 is outdated, as there was a new edition issued in 2012;

- the term "algorism" is incorrect, and should be replaced by "algorithm" throughout the manuscript;

- as per VIM 2012, the thousands separator shall not be used, so please remove the comma separator throughout the manuscript (e.g. "72572 gas" or "72 572 gas" instead of "72,572 gas", which causes an ambiguity - is it a decimal comma or a thousands separator?);

- as per VIM 2012, there shall be a space between the value and the percent symbol (e.g. "90 %" instead of "90%);

- in section 3.4, especially in the first paragraph, it is not clear is the number of transactions indicated is yearly;

- in the same section 3.4, the cost per certificate could be estimated as ~15 USD, but how this compare to the typical overall certification costs?

- in the same section 3.4, the authors claim that using a private blockchain such as HF would reduce the costs, but which would be the typical cost in this case, as compared to the above ~15 USD/certificate?

Author Response

We wish to express our appreciation to the reviewer for his or her insightful comments, which have helped us significantly improve the paper.

4-1)

Comment

in line 26, VIM 2007 is outdated, as there was a new edition issued in 2012;.

Response

As you pointed out, the latest version of 2012 is cited in the manuscript.

4-2)

Comment

the term "algorism" is incorrect, and should be replaced by "algorithm" throughout the manuscript;

Response

We have changed the description to "Algorithm" throughout the manuscript as you indicated.

4-3)

Comment

as per VIM 2012, the thousands separator shall not be used, so please remove the comma separator throughout the manuscript (e.g. "72572 gas" or "72 572 gas" instead of "72,572 gas", which causes an ambiguity - is it a decimal comma or a thousands separator?);

Response

We have removed the comma separator throughout the manuscript.

4-4)

Comment

as per VIM 2012, there shall be a space between the value and the percent symbol (e.g. "90 %" instead of "90%);

Response

We have inserted a space between the number and the % throughout the manuscript.

4-5)

Comment

in section 3.4, especially in the first paragraph, it is not clear is the number of transactions indicated is yearly;

Response

We have added "per year" in section 3.4 to emphasize that the number of transactions indicates yearly (L. 262, 263 and 265).

4-6)

Comment

in the same section 3.4, the cost per certificate could be estimated as ~15 USD, but how this compare to the typical overall certification costs?

Response

The above estimation you pointed out is correct. The fee for issuing a calibration certificate in Japan is generally between $10~$30. For reference, as of 2022, the NMIJ's fee for issuing a certificate for calibration and testing service except jcss is 1300 yen for Japanese text and 2300 yen for English text (excluding tax). We have added above information in the main text (L. 276~278).

4-7)

Comment

in the same section 3.4, the authors claim that using a private blockchain such as HF would reduce the costs, but which would be the typical cost in this case, as compared to the above ~15 USD/certificate?

Response

We found it difficult to compare the costs of each private chain in general terms, as they are largely dependent on the specifications of each. On the other hand, it is possible to form a blockchain without asking for transaction fees in HF. However, in such cases, it is assumed that the participants in the HF channel are trustworthy persons/organizations, which would reduce tamper resistance of the system compared to private chains. There also exists the issue of a single point of failure. In other words, there are advantages of lower transaction fees and other disadvantages that need to be considered from both perspectives.